# *Paratanytarsus grimmii* (Chironomidae) Larvae in Drinking Water Distribution Systems: Impairment or Disaster?

Stefan Christopher [1,*], Ute Michels [1] and Günter Gunkel [2]

1   AquaLytis, 15745 Wildau, Germany
2   Inwert Institut for Biological Drinking Water Quality, 13465 Berlin, Germany
*   Correspondence: stefanchristopher@aqualytis.de

**Abstract:** The occurrence and mass development of macroinvertebrates in drinking water networks is a challenge for drinking water pipe management. A current problem is the development of larvae of the chironomid *Paratanytarsus grimmii* (mosquito), a so-called pest organism that can have severe impacts on water quality due to mass accumulation from parthenogenic reproduction, biofouling and other aesthetic reasons. The aim of this study was to develop a new practical method for determining larvae size classes and analyzing the growth of the larvae. Knowledge of the dimensions, life cycle and fertility of these larvae within drinking water networks is essential for any risk analysis and the development of *P. grimmii* pest-control strategies. A two-year study of *P. grimmii* in a drinking water distribution system in Northern Germany was conducted, and *P. grimmii* population dynamics are presented. The parthenogenetic reproduction of *P. grimmii* without any pharate females (facultative flying stage) within the drinking water distribution system was proofed. In 2020 and 2021, five generations of *P. grimmii* were observed per year, with a maximum abundance of 6350 ind. m$^{-3}$. Mass accumulation occurred in the late-summer/autumn period.

**Keywords:** drinking water pipe inhabitants; macroinvertebrates; biofouling; drinking water quality; life cycle; larvae stages; drinking water pipe management

## 1. Introduction

*Paratanytarsus grimmii* (Schneider, 1885) [1] is a mosquito (Diptera, Chironomidae; Figure 1), the larvae of which were first recorded as a pest species in water distribution systems in 1941 [2]. Today, the species is known to be widely spread in Europe, South America, North America, Asia and Oceania [2–6].

*Paratanytarsus grimmii* is a triploid chironomid [2,3,7] that reproduces parthenogenetically (i.e., virgin reproduction) without any occurrence of males [7,8]. Reproduction takes place in the pupal stage of female animals (the so-called pre-imago stage) or pharate females (the facultative flying stage) without free-living mosquitoes. In drinking water distribution systems (DWDSs), pharate females (flying females that have not yet laid eggs) cannot occur, because drinking water networks are free from air space.

*Paratanytarsus grimmii* larvae build silken living tubes that are made from different materials, depending on availability (mainly sand, silt particles or fine detritus), which is directly connected to a substrate (e.g., the wall of a drinking water pipe, as shown in Figure 2). The living tubes can be up to 5 cm long and are irregularly twisted. The larvae lengthen the living tubes as they grow.

The life cycle comprises the following developmental stages: eggs → larva (four stages) → pupa → pre-imago (females only). The first larval stage involves individuals that are 0.6–0.8 mm long. Larvae then grow up to 6 mm in length and 0.5 mm in width.

The larval life cycle ends with metamorphosis (the pupa stage), which lasts only two days. These pupae then drift around in water flows. Laboratory-cultivated pre-imago *P. grimmii* in the pupa stage release eggs individually on long gelatinous filaments,

as described by Encina et al. [9], although this was not observed in pupae in drinking water networks. *Paratanytarsus grimmii* eggs are 260 μm long and 80 μm wide. A very conservative assumption is that each female lays an average of 40 eggs, based on data from laboratory experiments [9]; data of life cycle and reproduction in DWDSs are not available, especially the number of generations and the fertility of the larvae, and the necessity of the appearance of pharate females has not yet been clarified.

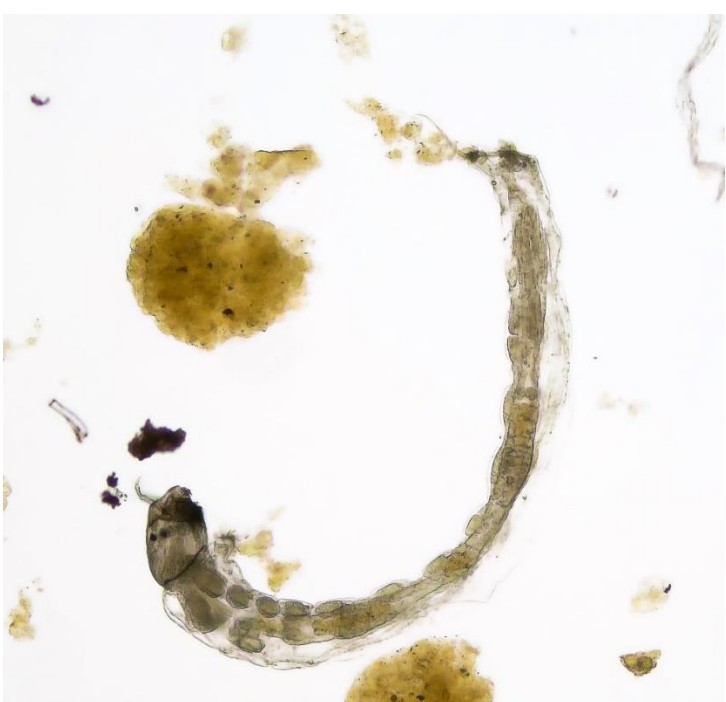

**Figure 1.** *Paratanytarsus grimmii* larva from a drinking water network, 5 mm long: Figure Michels©.

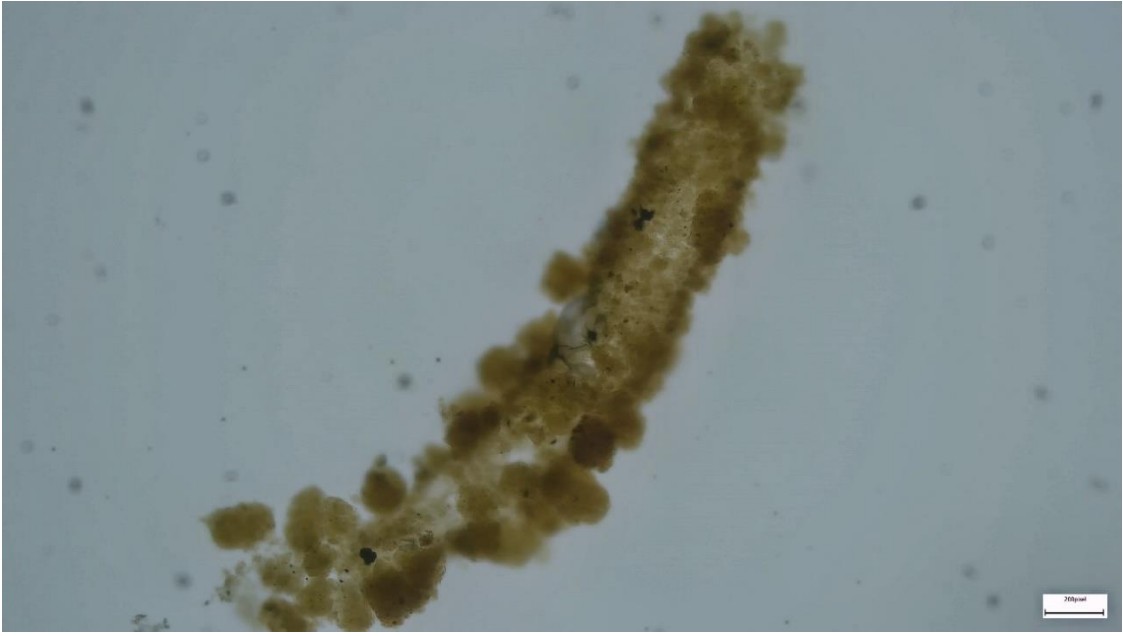

**Figure 2.** Living tube of *Paratanytarsus grimmii* from drinking water network. The living tubes reach a length of up to 5 cm, but they are damaged due to flushing out shear forces. The scale corresponds to 0.75 cm. Figure Michels©.

The control of *P. grimmii* occurrence in drinking water distribution systems must be a focus of any drinking water pipe management. The WHO [10] points out that the "Large invertebrate populations also indicate high levels of organic material that may give rise to other water quality issues, such as microbial growth. ". The German drinking water technical rule gives the key value of a low two-digit range for invertebrates >0.5 mm with groundwater as a raw water source [11].

Temperature is an important factor for *P. grimmii* mortality and life cycle completion. Olsen et al. [8] found that the highest larval mortality rate (80%) occurred at 12 °C, whereas the lowest (13%) was measured at 17 °C. In laboratory experiments at 15 °C, *P. grimmii* was observed to complete its life cycle in about 81 days [8].

In natural environments, larvae feed on organic detritus by the uptake of settled fine particles, which sink down into the living tubes, whereas in drinking water distribution systems (DWDSs), the available food consists of iron- and manganese-oxidizing bacteria and in situ flocculation enriched with attached bacteria. Thus, we must assume that *P. grimmii* larvae prefer lenitic conditions.

Data for the life cycle of *P. grimmii* in drinking water distribution networks are not yet available, but we assumed that the life cycle must be different from those of natural environments due to the different environmental factors and food sources. The aim of this paper is to provide the missing data about the life cycle and fertility of *P. grimmii* in DWDSs as well as to fill the lack of a practical method for analyzing larvae stages with the development of an adapted drinking water pipe management system to control *P. grimmii* development. New results are presented on the reproduction of the chironomid larvae in drinking water distribution networks, in particular whether the larvae can persist long term in hermetically sealed DWDSs. The analysis of the individual larval stages during drinking water pipe flushing records the mobility of larvae in DWDSs, and data on the reproduction and fertility of the larvae in the TW net are collected by means of 2-year monitoring.

The occurrence of *P. grimmii* in water distribution systems does not seem to pose a direct threat to public health, although some indirect impacts have been observed [12]. One problem with the occurrence of *P. grimmii* is biofouling [13], i.e., the risk of the promotion of harmful microbes during the fouling of pipe walls. There are also aesthetic issues, i.e., when chironomid larvae are visible in the house connection water filters or in the tap.

## 2. Materials and Methods

### 2.1. Study Site

The studied DWDS is located in Northern Germany, where the first records of *P. grimmii* occurred in 2019. Water supplies come from groundwater that has undergone advanced water treatments: (1) an aeration step and a flocculator with partial softening with calcium hydroxide and polyacrylamide, (2) a gravel/Jura lime filter for deferrization and demanganization, (3) a Jura lime filter for ammonium degradation, and (4) a reaction chamber for pH value adjustment; the complete water treatment occurs in closed factory halls with open filters, and no chlorination of the water is performed. The characteristics of the studied drinking water are presented in Table 1.

*Paratanytarsus grimmii* were obtained from samples in 2021 using water hydrant flushing, $CO_2$ flushing or a combination of $CO_2$ and umbrella flushing during the monitoring of the studied DWDS, in which the pipes were flushed for chironomid control (for further details, see [14]). The pipe-flushing outflow was filtered using a low-pressure high-flow-through stainless steel filter with a mesh size of 100 μm, which allowed for the collection of fine organisms without damaging them [15].

### 2.2. Paratanytarsus grimmii Analysis

Based on the work of Langton et al. [5], larval stages were determined according to head capsule length. Since some readings did not fall into Langton's pattern, the measurements were adjusted, especially for larval stages L1 and L3.

**Table 1.** The characteristics of the drinking water from the studied network in Northern Germany, which was populated with *Paratanytarsus grimmii*.

| Parameter | DWDS Sample |
|---|---|
| Raw water source | Groundwater |
| Drinking water treatment | Aeration, flocculation and filtration |
| Chlorination | None |
| Temperature at waterworks outlet (°C) | 9.9–11.1 |
| Temperature in DWDS: Winter period Summer | 8–10 °C |
| period | 16–22 °C |
| Conductivity ($\mu$S cm$^{-1}$ at 25 °C) | 255 |
| pH | 7.9–8.1 |
| Turbidity (NTU) | 0.2 |
| Color (SAC 436 nm) | 0.48 |
| Total Fe (mg L$^{-1}$) | 0.08 |
| TOC | 6.1 |

Head capsule length was measured from the anterior end of the antenna base to the outermost occipital border, and head capsule height was measured from above the eyes (Figure 3). Mouthparts (labrum, mandibles and mentum) were excluded from the measurements, but head capsule length was included within the total body length. Only individuals that were laid on their sides were measured. Additional measurements of the head capsule and thoracal height were carried out to establish the mesh size that would catch L1 larvae (e.g., in drinking water network filters).

In the years 2020–2022, monthly monitoring of the abundance of *P. grimmii* was conducted using six hydrants distributed over the drinking water network.

The *P. grimmii* larvae studied were flushed out of the drinking water distribution system of the water work, and the pathway of introduction of the *P. grimmii* larvae is unknown (e.g., contaminated hydrants standpipes, contaminated hoses of the fire brigade, pipe network work).

The *P. grimmii* samples were fixed with ethanol (96%) and analyzed and measured in the laboratory under optical magnification. The chironomid larvae were categorized based on their morphological characteristics [16,17], and the results were validated by DNA barcoding, which was conducted by AIM (Advanced Identification Methods GmbH, Leipzig, Germany).

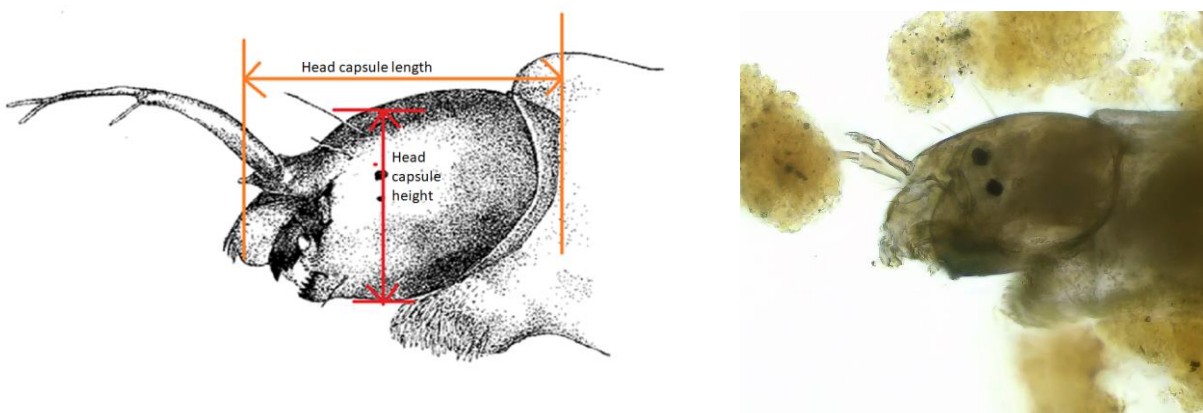

**Figure 3.** A lateral view of the head of a mosquito larvae (*Tanytarsus* sp.). Drawing modified from Carew [18], right figure microscopic view, Michels©.

*Paratanytarsus grimmii* were picked from different samples, positioned on microscope slides (in a drop of tap water) and covered with coverslips. All measurements were conducted using an inverted microscope (Olympus CKX41 with WHB10X-H/20 ocular and

Cach N 10×/0.25 PhP ∞/−/FN22 objective, magnification 100×) shortly after preparing the microscope slides. The ocular scale measured 987 μm in total.

The developed method for the detection of *P. grimmii* larvae included the following steps:

(1) *Paratanytarsus grimmii* larvae fixed in alcohol were exposed on a microscope slide with a drop of tap water and measured under an inverted microscope or, for routine analysis, with a binocular with ocular scale.

(2) The total length (in 0.1 mm steps) was determined, from the beginning of the head capsule to the abdomen.

(3) Total body length was assigned to the larval stages by the following length classes: L1 larva = < 1.1 mm, L2 larva = 1.1–2.1 mm, L3 larva = 2.1–3.2 mm and L4 larva = > 3.5 mm (see Table 2).

**Table 2.** The head capsule lengths of the *Paratanytarsus grimmii* larval stages.

| | Head Capsule Length of *P. grimmii* (mm) | | | | |
|---|---|---|---|---|---|
| **Larval Stage** | **10th Percentile** | **25th Percentile** | **Median** | **75th Percentile** | **90th Percentile** |
| L1 (*n* = 110) | 0.08 | 0.08 | 0.08 | 0.09 | 0.09 |
| L2 (*n* = 66) | 0.11 | 0.11 | 0.12 | 0.12 | 0.13 |
| L3 (*n* = 290) | 0.18 | 0.19 | 0.20 | 2.21 | 0.22 |
| L4 (*n* = 186) | 0.29 | 0.30 | 0.31 | 0.33 | 0.35 |

## 3. Results

### 3.1. Biometric Parameter of Paratanytarsus grimmii Larvae

The head capsule length of the *P. grimmii* was used to differentiate between the larval stages using the median lengths of 0.08 mm (L1 larvae), 0.12 mm (L2 larvae), 0.20 mm (L3 larvae) and 0.31 mm (L4 larvae) (Table 3).

**Table 3.** The total body lengths of the *Paratanytarsus grimmii* larval stages.

| | Total Body Length of *P. grimmii* (mm) | | | | |
|---|---|---|---|---|---|
| **Larval Stage** | **10th Percentile** | **25th Percentile** | **Median** | **75th Percentile** | **90th Percentile** |
| L1 (*n* = 110) | 0.58 | 0.65 | 0.76 | 0.93 | 1.04 |
| L2 (*n* = 66) | 1.19 | 1.36 | 1.73 | 1.87 | 2.05 |
| L3 (*n* = 290) | 2.33 | 2.51 | 2.69 | 2.95 | 3.18 |
| L4 (*n* = 186) | 3.47 | 3.90 | 4.30 | 4.71 | 5.46 |

The total length was also determined. Larval stage L1 had a median length of 0.8 mm (with 0.6 and 1.0 mm as the 10th and 90th percentiles, respectively), larval stage L2 had a median length of 1.7 mm (with 1.2 and 2.1 mm as the 10th and 90th percentiles, respectively), larval stage L3 had a median length of 2.7 mm (with 2.3 and 3.2 mm as the 10th and 90th percentiles, respectively), and larval stage L4 had a median length of 4.3 mm (with 3.5 and 5.3 mm as the 10th and 90th percentiles, respectively) (Table 2).

Correlations between the total body lengths enabled us to identify the larval stages. The body lengths of the different larval stages did not overlap when we considered the 10th and 90th percentiles (Figure 3).

The correlations between the head capsule lengths and the total body lengths clearly demonstrated some overlaps in body size (i.e., there were some data that did not correspond well for head capsule lengths of 0.15 mm and 0.25 mm). For these data, it became difficult to distinguish between larval stages just using body lengths. L1 and L2 larvae overlapped by <7%, L2 and L3 larvae overlapped by <3%, and L3 and L4 larvae overlapped by <5%.

The body lengths ranged from 0.54 to 6.08 mm across all stages (Figure 4).

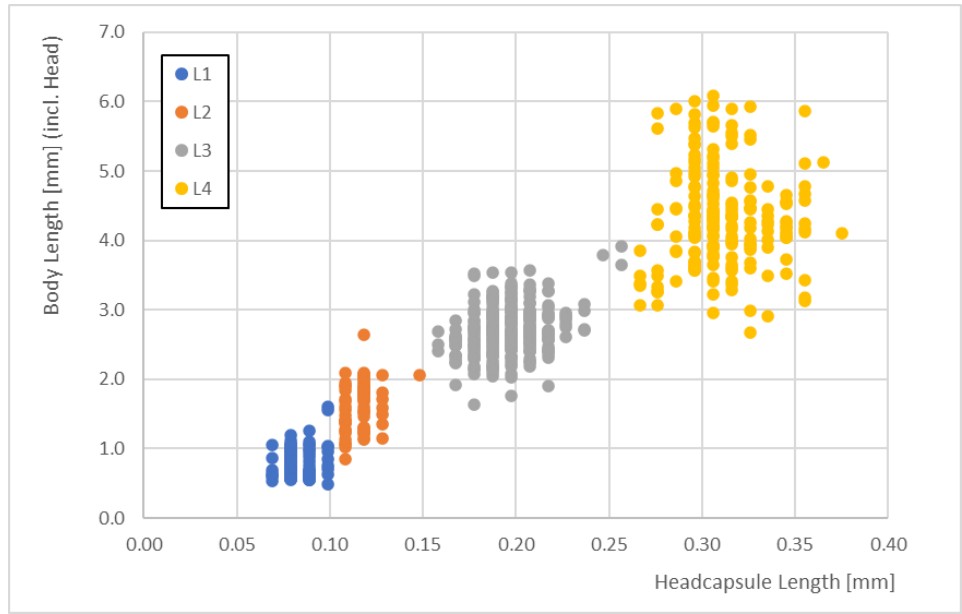

**Figure 4.** The correlations between head capsule length and total body length for the four larval stages (L1 to L4) of *Paratanytarsus grimmii* (*n* = 652).

In contrast, the head capsule heights did not offer better correlations with the body lengths. Figure 5 shows the distribution of head capsule heights for the different larval stages, which was ≤100 μm for L1 and L2. Across all stages, the distribution ranged from 0.05 to 0.25 mm. Larval stages L1 and L2 strongly overlapped at a head capsule height of 0.07 mm. Additionally, some L3 and L4 data demonstrated strong scattering.

Thus, for practical use and routine analyses, the correlations between head capsule lengths and total body lengths could be applied, and the obtained size classes for the larval stages could be a tool for *P. grimmii* population analysis (Table 2).

### 3.2. Abundance of Paratanytarsus grimmii in a Drinking Water Distribution System

Our three years of monitoring *P. grimmii* development in the DWDS in Northern Germany clearly showed oscillations in their abundance, with the highest values occurring in late summer/autumn. In the winter/spring period, larvae density dropped to less than 40 individuals per m$^{-3}$ of water; however, in late summer, abundance increased up to 6.350 larvae per m$^{-3}$ of water (Figure 6).

Within a few years *P. grimmii* spread over large parts of the drinking water distribution network, even though *P. grimmii* larvae usually stay within the depths of the living tubes. This protects them from being ejected by high flow conditions (e.g., pipe network flushing). After hatching, however, the young larvae (L1) can crawl for short distances and find new places to build their living tubes. Analyses of *P. grimmii* larvae stage during pipe flushing clearly pointed out that L1 larvae can drift during this first period; drifting was also observed of the pupae, which are not fixed to the pipe wall.

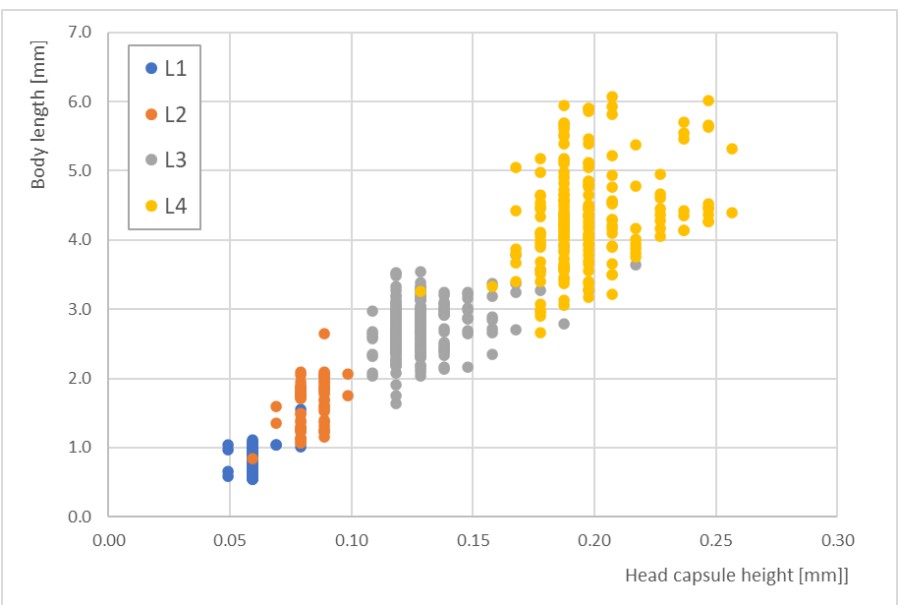

**Figure 5.** The correlations between head capsule height and total body length for the four larval stages (L1 to L4) of *Paratanytarsus grimmii* (*n* = 565).

Thus, the spreading of *P. grimmii* larvae and pupae occurs, supported by pipe flushing and similar activities involving high flow rates. This is a challenge for controlling the populations of *P. grimmii* in drinking water distribution systems.

It should be noted that the discharge rate of *P. grimmii* under standard pipe monitoring conditions amounts to 10–30% for high densities, because the larvae are fixed at the bottom of their living tubes. The almost complete ejection of *P. grimmii* has been achieved using a combination of $CO_2$ flushing and umbrella flushing [19,20]. Based on these data, we could calculate the hydrant sampling efficiency for *P. grimmii*. Considering the efficiency of hydrant monitoring, we can calculate a *P. grimmii* abundance of about 20,000 larvae per $m^3$; this corresponds to a population density of 1300 larvae per $m^2$ pipe wall. The biomass of the *P. grimmii* larvae amounted to 113 mg $m^{-3}$, and the abundance and biomass must be evaluated as biofouling.

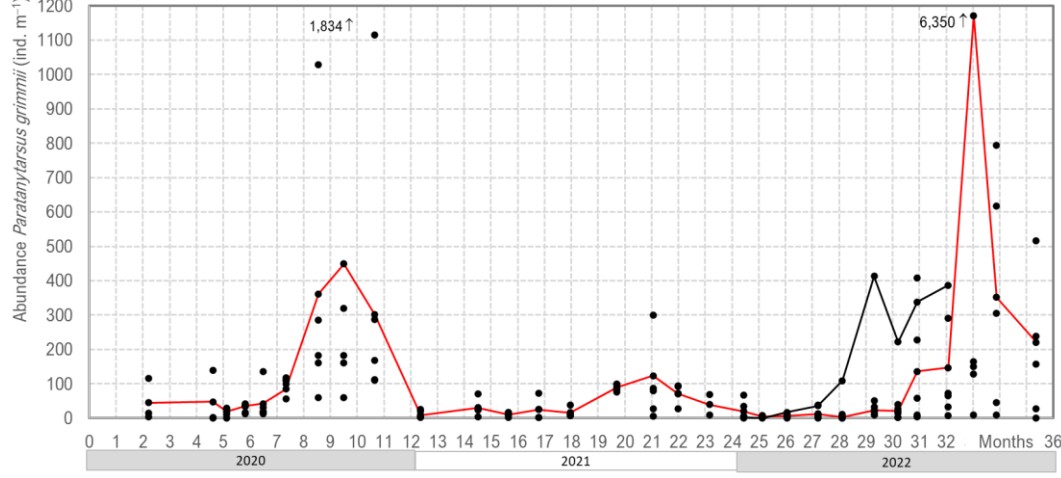

**Figure 6.** Our monitoring of the abundance of *Paratanytarsus grimmii* in a drinking water distribution system in Northern Germany (the red line shows the mean of six hydrant water samples, and the black points represent the individual monitoring data).

This abundance of *P. grimmii* must be evaluated as very high considering the risk of the aesthetic and microbial impact of the drinking water.

Comparable abundances have been reported in the literature, for activated carbon filters with 3800 individuals per $m^2$ in the United Kingdom [7], thousands of larvae in a Danish drinking water well [21] and mean densities in drinking water networks of 140–560 individuals per 2.3 $m^3$ in the mid-western US [3].

In the DWDS, coliform bacteria number is increased (Figure 7), and a tendency is given in which the occurrence of coliforms (*Serratia fonticola*) is supported by *P. grimmii*, but no significant correlation exists ($R^2 = 0.02$).

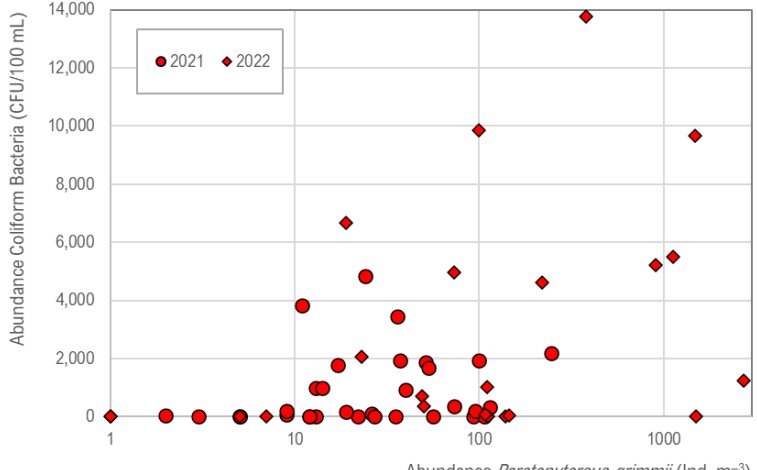

**Figure 7.** Correlation of *Paratanytarsus grimmii* abundance (ind. $m^{-3}$) and coliform bacteria (CFU 100 $mL^{-1}$), with a tendency of co-existence. The number of coliform bacteria is determined with pipe flushing of 1 m $s^{-1}$ and does not correspond to normal conditions of drinking water supply.

### 3.3. Population Dynamic of Paratanytarsus grimmii

Our population dynamics analysis of the *P. grimmii* population in the studied drinking water network based on the presented larvae size classes showed five generations per year (F1 to F5), with mass accumulations occurring in late summer/autumn. The occurrence of pupae could not be used as an indicator for reproduction and population growth, because the life span of pupae is only two days, so the observation of pupae can only be achieved through daily monitoring. Thus, the body lengths of the *P. grimmii* larvae were used to monitor the growth and reproduction rates of the population in the drinking water distribution system. Both parameters (the relative amounts of larvae in the different stages and the total number of larvae per $m^3$ of pipe volume) were considered in our growth analysis (Figure 8).

The overwintering of the larvae (F0 generation) led to a severe decrease in their abundance. This seemed to demonstrate the increased mortality of *P. grimmii* larvae with decreasing temperature. In spring, several generations occurred, but they also had high mortality rates; therefore, there was only a slight increase in abundance (F1 to F3 generations). The F4 generation was first observed in August, and the F5 generation was observed in October, with strong increases in abundance.

The mean population during the summer period was 70 days (standard deviation: 16 days), whereas in winter period, the life span of *P. grimmii* increased up to 152 days, demonstrating the significant effects of winter temperatures.

The mean abundance increased from 10–40 individuals per $m^{-3}$ in spring to 400–1100 individuals per $m^{-3}$ in autumn, with a maximum of 6350 individuals per $m^{-3}$. It was interesting to note that the maximum abundance differed from 123 to 6350 individuals per $m^{-3}$ throughout the investigation period. This meant that every year, a new population occurred with a mass accumulation, starting with only a few overwintering larvae.

The mass development of *P. grimmii* in DWDSs is triggered by environmental factors, i.e., temperature, flow rate and available food.

**Figure 8.** The population dynamics of *Paratanytarsus grimmii* in a drinking water distribution system in Northern Germany, showing the relative proportions of the population in larval stages L1 to L4 (top panel) and the total number of larvae in all stages (bottom panel) (in ind. m$^{-3}$). The areas of the dots correspond to the relative proportions of the total number of larvae. The colors represent the different generations.

## 4. Discussion

*Paratanytarsus grimmii* has long been known to be a pipe inhabitant in many countries across the northern hemisphere throughout the last century. However, recently, *P. grimmii* populations have occurred in DWDSs that are fed with groundwater that has undergone state-of-the-art drinking water treatments. This has presented a new and unexpected situation that needs to be evaluated critically, because mosquito larvae can reach high population densities (up to >6000 larvae per m$^{-3}$ of hydrant flushing water, respectively > 1000 larvae per m$^2$ pipe wall) within pipes and lead to biofouling [11]. There are also aesthetic impacts and a decreased acceptance of drinking water when mosquito larvae are visible (e.g., on filters in household taps).

Additional intestinal biomes from macroinvertebrates can support the development, survival and spreading of harmful bacteria by acting as carriers through feces deposition [22, 23]. Additionally, the mucus of eggs can serve as a substrate for bacterial growth [24,25]. Up to now, only limited data have been available. Studies on the biome of *Asellus aquaticus* have proved the occurrence of harmful bacteria, such as *Pseudomonas* sp. and *Aeromonas hydrophila* [26]. Christensen [27] reported that *E. coli* bacteria settled in the intestines of *Asellus aquaticus*, but the amount was too low and insignificant to qualify as drinking water contamination.

The risk of *P. grimmii* accumulation in DWDSs mainly comes from biofouling and related problems, such as microbe development (e.g., coliform bacteria and *Serratia fonticola*). It is assumed that living tubes that are constructed from the available materials in pipes provide good substrates for microbe development. Larvae themselves can also be used as substrates for microbe development (i.e., attached to their surfaces or as biomes in their intestines), but up to now, insufficient data have been available to support this.

Open urban wells that are fed with drinking water offer preferable habitats for *P. grimmii* [21] and can be a source of spreading. The introduction of mosquitos into DWDSs can only occur through pharate females and not hermetically closed drinking water treatment

systems (filters, tanks, etc.) [7]. Flying pharate females deposit eggs into the water, where no males or flying female mosquitos are needed for further development due to their parthenogenetic reproduction.

According to Langton et al. [28], the average body length of L4 larvae is 3–4 mm (body lengths in other stages were not measured). Our data showed that some L4 larvae had body lengths of over 5 mm. The median length (including the head) was 4.3 mm, with 25th and 75th percentiles of 3.9 and 4.7 mm, respectively.

The presented analysis of *P. grimmii* larval stages and body lengths provided sufficient size classifications, with overlaps of <5% (except for L1 and L2 larvae). Thus, the body lengths of *P. grimmii* larvae could be used as a good and applicable tool for population dynamics analysis.

Monitoring can be conducted via hydrant sampling (with a high flow rate of about $1 \text{ m s}^{-1}$) or pipe flushing using advanced technologies [18,29]. The efficiency of hydrant monitoring is low (in the range of 10% (low abundance) to 30% (mass accumulation)), which corresponds to the insufficient pipe flushing success of conventional flushing methods, such as water or air/water.

As well as abundance, population structure is significant for all DWDS management and pipe flushing activities. Our data clearly showed that L1 larvae and pupae were mobile (there are no available data for eggs); however, the observed deposition of singular eggs without long gelatinous filaments by pre-imago individuals results in eggs drifting around in the pipe water flows.

These mobile phases in the life cycle of *P. grimmii* (eggs, L1 larvae and pupae) support spreading via water flows within drinking water networks, mainly through routine pipe flushing. Considering mosquito larvae abundance and development, the optimum time periods for flushing activities without supporting the spread of the larvae would be in spring/early summer, when low abundance of *P. grimmii* and periods without reproduction (i.e., no pupae, eggs or L1 larvae) occur. The optimal conditions can be determined through pipe monitoring and population dynamics analyses based on larvae growth data.

Overall, we found that total body length was a sufficient parameter to determine larval stages. Our monthly monitoring over nearly three years enabled a population dynamics analysis of a *P. grimmii* population in a DWDS in Northern Germany. The occurrence of five generations per year demonstrated the very high reproduction potential of *P. grimmii*. The observed maximum of 6350 mosquito larvae per $\text{m}^3$ of pipe (with a mean of 450 mosquito larvae per $\text{m}^3$ of the drinking water network) was reached in late summer, whereas the population density in spring was in the low two-digit range.

## 5. Conclusions

*Paratanytarsus grimmii* is a well-known mosquito larva in water distribution systems all over the world, a triploid chironomid that reproduces partheno-genetically without any occurrence of males. Long-term population development can also occur within DWDSs without pharate females (the facultative flying stage).

*Paratanytarsus grimmii* larvae grow up to 5 mm and build silken living tubes that are directly connected to the wall of the drinking water pipe; the living tubes can be up to 5 cm long.

In the first larvae stage, *P. grimmii* can crawl for short distances, but they have also been observed to drift during this first period; therefore, the risk of displacement in drinking water networks applies to L1 larvae and the free-floating pupae, supported by pipe flushing and similar activities involving high flow rates.

A more practical method for the determination of *P. grimmii* larvae stages L1 to L4 has been developed, based on determination of the total body length with individual larvae size classes.

Three years of monitoring *P. grimmii* development clearly show oscillations in their abundance. In the winter/spring period, larvae density drops to less than 40 individuals per $\text{m}^{-3}$ of water; however, in late summer, abundance increases up to 6350 larvae per $\text{m}^{-3}$

of water. Considering the efficiency of hydrant monitoring, we can calculate a *P. grimmii* abundance of about 20,000 larvae per m$^3$, this corresponds to a population density of 1300 larvae per m$^2$ pipe wall, and this abundance must be evaluated as biofouling.

Temperature is an important factor for *P. grimmii* mortality and life cycle completion. Our population dynamics analysis of the *P. grimmii* population in the studied drinking water network based on the presented larvae size classes shows five generations per year. The mean population during the summer period is 70 days, whereas in winter period, the life span of *P. grimmii* increases up to 152 days.

Data on the drinking water quality by hydrant sampling indicate an increased risk of the development of coliform bacteria (*Serratia fonticola*) in the case of mass development of *P. grimmii* larvae.

**Author Contributions:** S.C. carried out data curation and original draft preparation, U.M. directed the biological analyses and did project administration, G.G. reviewed and edited the paper. All authors have read and agreed to the published version of the manuscript.

**Funding:** This research received no esternal funding.

**Institutional Review Board Statement:** Not applicable.

**Informed Consent Statement:** Not applicable.

**Data Availability Statement:** Data are unavailable due to privity character of water quality supply.

**Acknowledgments:** This work was made possible by the support of various drinking water supplier who must be anonymous due to the sensitive drinking water data. Thanks are due for the provision of resources and openness to systematic investigations in the drinking water networks.

**Conflicts of Interest:** The authors declare no conflict of interest.

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
