# Peer review of "Paratanytarsus grimmii (Chironomidae) Larvae in Drinking Water Distribution Systems: Impairment or Disaster?"

_water, doi:10.3390/w15030377_

Round 1

Reviewer 1 Report

The topic of the paper is interesting and suitable for the journal. The authors study the occurrence of the larvae of Paratanytarsus grimmii (mosquito) in drinking water distribution systems of North Germany during 2020 and 2021. In this study, the authors also want to develop a practical method for determining larvae size classes and applying it to analyse the P. grimmii population dynamics in drinking water networks.

After reading the paper, minor revision is required prior to possible publication. the following issues should be addressed:

Lines 20-21. The aim was to develop a new practicable method for determining larvae size classes….

A section (or sub-section) explaining the steps in detail of the proposed method should appear in the manuscript, so that anyone interested in applying the proposed method has no doubt (nor to interpret how it should be done).

Lines 87 -90. Please, provide additional (and detailed) information on the groundwater treatment steps:

·       type of sand filters, granulometry, thickness, open/closed vessels,…

·       Which type of oxidation? (ozone, permanganate, contact time....)

Larvae Source. If the raw water source is groundwater, and it goes through the water-treatments... how do the eggs and larvae originate? If there are open water transport or treatments that allow the deposition of eggs/larvae... it should be clearly indicated.

Table 1. One of the treatment steps is Oxidation according to the text in line 88 but this water-treatment is not in the table.

Figures 4 and 5. Is there any reason not to include the correlation equation, the correlation coefficient (R2), and the RMSE of a linear fit (together with the experimental data already shown) ?

In section 6.2, the authors present the data that they have measured... but it would be necessary to discuss these numbers in comparison with other studies (is this abundance, similar, high or low compared to other drinking water networks) as well as arguing about critical limits... i.e., when would the abundance of P. grimmii be problematic, or worrying? ... From what number of individuals per cubic meter do we have to worry about?

Figure 7. A small detail in the legend of x-labels:

Dez -> Dec

Mai -> May

Okt -> Oct

Conclusion Section is missing. It would be nice to see a Conclusion Section with the take-home messages.

Line 294. There is number “1” before this reference [10]. Please remove that “1”.

Line 296. Reference number [15] between [10] and [11]. Please re-organize.  

Author Response

Reply to reviewer 1 L20 – 21, I have inserted a short summary of the proposed method step by step (L149 – 153) L 87-90: raw water treatment is given more precise (L101 – 105) Larvae source is explained (L145 – 148) In fig 4 and 5 show first of all the variance of the L1 to L4 larvae stage in head capsule length and body length, the statistical data are given in Table 2 and 3. I think it is not useful to insert the regression line resp. the correlation coefficient for all data because the growth of the larvae is not focus, but only the size classes of different larvae stages. Fig. 7 changed I inserted a Conclusion Section L407-4035 References are corrected.

Reviewer 2 Report

Dear authors,

Your manuscript is generally well written, but you have spelling errors that must be corrected. All comments were included in the manuscript. Please also review the list of articles - some items are not edited following the journal's requirements.

Author Response

Reply to reviewer 2

Small changes marked in the text, all done: L30, L34, L37, L43, L59, L64, L91, L96

L 83: I have changed the text to point out the significance of biofouling

Fig. 2: the scale is a pixel scale, I inserted the dimension in the legend

L97- L100: moved to section P. grimmii analyses

Small changes in the text, all done: L103, L110, L114, L121, L123, L124

L 135-136 deleted

Small changes done, L 150, L 155, L 158, L162

L 161: the years of the monitoring are given in the methods (L 145)

Fig. 6 changed

Small changes done, L183, L186, L202, L224

The English spell check was done by MDPI editing service.

Reviewer 3 Report

General concept comments:

The introduction should be rewritten and organized more logically to better show the importance and significance of this research.

The experimental design of this study is very interesting. Nevertheless, the residual disinfectant which is available in treated water before being distributed, should be able to kill any pathogens.

Abstract

Should be re-phrase: short problem statement, aim of study, short description on methodology and major outcomes

Introduction:

The introduction is not well organized. The authors should review the advance, challenges and unsolved problems in this field and state the importance and novelty of this study. This is important to show gap in knowledge.

2. it is suggested to use set of consistent keywords (in abstract, intro ,and conclusion)

Materials and research method

1. for this section - Missing Reference on methods used

2. This section is more on reporting style.  

Result and Discussion

1.    This section is more on reporting style. Previous study should be included and any related theory should be properly cited

2.    Discussion in this section must reflect the topic.

3.    Impairment or disaster – this should be well answered in this section

4.    The description in this section is still limited to the characteristic of the pathogen and there was no graph to see the correlation between this microbes can fouling within the distribution

5. The authors should clearly classify relevant indicators. (rephrase the aim of study) and add conclusion to tally with am.

6. Data Presentation of this manuscript, must be improved.

Conclusion

Must  be included and tally with objectives

Author Response

Reviewer 3

General concept comments:

The experimental design: No disinfectant were used in the water treatment

Abstract: revised and shortened

Introduction: reorganized and supplemented

The introduction is not well organized. The authors should review the advance, challenges and unsolved problems in this field and state the importance and novelty of this study. This is important to show gap in knowledge.

Keywords: changed

Materials and research method

  1. references for sampling and monitoring are [14] and [15], the data of the drinking water distribution system have to be anonym
  2. This section is supplemented and revised

Result and Discussion

  1. This section is more on reporting style. Previous study should be included and any related theory should be properly cited –

This is done in line L203-L206

  1. Discussion in this section must reflect the topic / 3. Impairment or disaster – this should be well answered in this section

This is done by lines L201-L202

  1. The description in this section is still limited to the characteristic of the pathogen and there was no graph to see the correlation between this microbes can fouling within the distribution

I have inserted the figure 7 (new) with the tendency of P. grimmii and coliform bacteria

  1. The authors should clearly classify relevant indicators. (rephrase the aim of study) and add conclusion to tally with am. The method is summarized in L 261-268
  2. Data Presentation of this manuscript, must be improved.

The data presentation is given in fig 4 and 5 with the statistic in Table 2 and 3, we cannot give any statistic correlation for the data of Fig. 4 and 5, because we analyzed size classes to distinguish the larvae stage.

Conclusion

Must  be included and tally with objectives Conclusion is inserted.

Round 2

Reviewer 3 Report

1. Abstract is not well written : Too long introduction and methodology appear towards the end of paragraph. In this section, start with problem statement, followed by the aim,the discuss on methology. The final finding should be the outcome of the study.

2. still no gap in knowledge..

3. The end of the paragraph of the introduction,   gap and  objectives of the study should be included.,

Author Response

Response to reviewer 3

Abstract is rewritten as you recommended (problem / aim / method / findings)

Now the gap of the paper is focused in the abstract (line L17 – L18)

Introduction Knowledge: gap is given in L205 – L218

Introduction is re-organized (L219 -L224)

Gap and objectives are given in introduction L234 - L244

Results: Own results and gap  is pointed out in Chap. 3.2 (L445 – L454)

In Conclusion gap of the paper in complemented by L612-613 and L635 - L637